# Mechanism underlying delayed rectifying in human voltage-mediated activation Eag2 channel

Mingfeng Zhang [1,2,3], Yuanyue Shan[1,2,3] & Duanqing Pei [2] ✉

The transmembrane voltage gradient is a general physico-chemical cue that regulates diverse biological function through voltage-gated ion channels. How voltage sensing mediates ion flows remains unknown at the molecular level. Here, we report six conformations of the human Eag2 (hEag2) ranging from closed, pre-open, open, and pore dilation but non-conducting states captured by cryo-electron microscopy (cryo-EM). These multiple states illuminate dynamics of the selectivity filter and ion permeation pathway with delayed rectifier properties and Cole-Moore effect at the atomic level. Mechanistically, a short S4-S5 linker is coupled with the constrict sites to mediate voltage transducing in a non-domain-swapped configuration, resulting transitions for constrict sites of F464 and Q472 from gating to open state stabilizing for voltage energy transduction. Meanwhile, an additional potassium ion occupied at positions S6 confers the delayed rectifier property and Cole-Moore effects. These results provide insight into voltage transducing and potassium current across membrane, and shed light on the long-sought Cole-Moore effects.

Cell fate decisions often entail intricate interactions between intrinsic mechanisms such as chromatin dynamics and extrinsic signals. While intrinsic pathways tend to be hardwired and well-programmed, the extrinsic signals are far more diverse and much less well-understood mechanistically. Classic cytokine/growth factors such as BMP, SHH, FGF, and WNT families have been shown to play critical roles in many developmental, physiological, and pathological pathways. Less appreciated, but emerging as critical, are physical signals, mechanical or electric, that regulate cell fate control through mechanosensing or voltage sensing mechanisms[1]. Voltage or electric signals can be detected at the cellular level by cell surface proteins containing voltage sensor domains or VSDs. Voltage-gated potassium (Kv), sodium (Nav), and calcium (Cav) channels all contain VSDs made of four helices, S1–4, with a series of positively charged amino acids (Arginine or lysine) lining one face of S4 that sense electrical field. In domain-swapped Kv channels, VSDs receive electromechanical signals and transduce them to the pore through the movement of VSDs and S4–S5 linkers, leading

to channel opening and propagation of electric current or action potential in the nervous system. However, how non-domain-swapped channels sense voltage and transmit electric signals remains less well understood.

Mammalian homologs of *Drosophila melanogaster ether go go* are non-domain-swapped Kv channels. Curiously, rat EAG (rEAG), was the first to show classic transforming activity in NIH3T3 cells[2], suggesting that it is capable of converting a somatic fate to a malignant one. Subsequent studies involving both members of EAG1 and 2 provided convincing evidence that both are oncogenic through diverse mechanisms involving cell cycle, volume, and signaling downstream to p38/MAPK pathways. Furthermore, the mutation in human *Eag2* can cause epileptic encephalopathy and autism diseases. Despite these diverse pathophysiological roles, the structural basis of the EAG channel function remains largely unknown.

Here, we report six structures with near-atomic resolution captured by cryo-EM, representing human Eag2 (hEag2) in closed, pre-

[1]Fudan University, 200433 Shanghai, China. [2]Laboratory of Cell Fate Control, School of Life Sciences, Westlake University, 310000 Hangzhou, China. [3]These authors contributed equally: Mingfeng Zhang, Yuanyue Shan. ✉e-mail: peiduanqing@westlake.edu.cn

open, open, and pore dilation but non-conducting states. Mechanistic insights from these structures may lead to a better understanding of voltage sensing and transduction in both electrophysiology of nerve systems as well as cell fate control during tumorigenesis.

## Results

### Structure determination of human potassium channel hEag2
We expressed full-length hEag2 fused with a green fluorescent protein (GFP) in HEK293T cells and characterized its electrophysiological properties and show that it is a voltage-activated delayed rectifier potassium channel with a half-activation membrane potential of −10.0 ± 3.5 mV (Fig. 1a, b). Interestingly, it appears to exhibit a typical Cole−Moore effect by holding the cell at increasing holding potentials, from −140 mV to the voltage of channel activation and stepping to the same depolarized voltage of +60 mV. The holding potential that produces half-maximal rates of activation is about −86.9 ± 15.2 mV, which is like the recording of rEag1 in CHO cells (Fig. 1c, d)[3,4].

To explore the mechanism of voltage activation, delayed rectifier, and Cole−Moore effect, we attempted to capture all potential discrete conformations in the depolarized voltage of 0 mV, which may be equivalent to the hEag2 protein in the buffer without any voltage stimuli. We purified the hEag2 in a 150 mM KCl environment with multiple bands based on SDS−PAGE, indicative of post-translational modifications (PTMs) and putative Hsp70 protein (Supplementary Fig. 1a, b). Since the endogenous calmodulin can bind to KCNH family proteins[5], the purified hEag2 may contain the calmodulin, such that the SEC peak seems to have no discernible monodispersed peak. We reconstitute the purified protein into liposome and recorded the typical voltage-activated delay rectified current, suggesting that the protein is functional and behaves similarly to native ones in the cell membrane (Supplementary Fig. 1c). We then subjected the purified protein to the standard cryo-EM workflow. After cryo-EM imaging and data processing (Supplementary Fig. 2), we isolated six distinct near-atomic resolution high-quality maps at an overall resolution of 3.5 (PDB:7YIH; EMD-33859), 3.5 (PDB:7YIG; EMD-33858), 3.4 (PDB:7YIF; EMD-33857), 3.4 (PDB:7YIE; EMD-33856), 3.5 (PDB:7YID; EMD-33855), 3.8 (PDB:7YIJ; EMD-33860) Å, respectively (Supplementary Figs. 2–5). We combined all particles from the six classes and got a worse (3.5 Å) map, suggesting there was significant conformational heterogeneity within the samples (Supplementary Fig. 2). These high-quality maps allowed us to build a near-atomic model of hEag2 at different conformations (Supplementary Figs. 6, 7, Table 1). Like rEag1 and hErg1[5,6], each hEag2 subunit contains an N-terminal PAS domain, voltage-sensor domain (VSD) formed from the S1–S4 transmembrane helices, pore domain formed from the S5-pore loop-S6 domain and the C-terminal CNBHD domain (Fig. 1h, i). The overall structure of hEag2 is identical to the rEag1 and hErg1 (Supplementary Fig. 8), with a "butterfly" shape. The channel has a three-layer architecture: the upper extracellular turret, transmembrane domains, and intracellular domains occupying 3D space with a size of 130 Å × 130 Å × 130 Å (Fig. 1e–g). The tetrameric hEag2 apparatus adopts noncanonical non-domain-swapped VSDs, with distinct pore domain configurations (Fig. 1j, k).

### hEag2 transitions between closed and open states
Six distinct configurations captured for hEag2 are primarily based on the pore dilation and ion distribution. Of which, two (PDB: 7YID and 7YIE) are closely related with a narrow ion conduction pore tentatively identified as in a closed state (Fig. 2a, b), one (PDB: 7YIH) with the distinctly extended intracellular gate as the open state (Fig. 2e), another (PDB: 7YIJ) with an extended intracellular site and one potassium ion occupied in the S4 as the open to closed transit state (we name it as pore dilation but the non-conducting state) (Figs. 2f, 3f), and the last two as pre-open state 1 (PDB: 7YIF) and pre-open state 2 (PDB: 7YIG), respectively, due to the dilation of ion conduction pore (Fig. 2c, d, g).

The overall Cα atoms and the side chain of the two closed states are almost identical (Fig. 2a, b, g), while the differences are density peaks of the water molecule and potassium ion in S0 (Fig. 3a, b), indicating that the water and potassium in S0 layer are dynamic in the closed states. The difference between the pre-open state 1 and pre-open state 2 is the side chain direction of Y460 (Figs. 2g, 4, and 5a). In the pre-open state1, the side chain of Y460s inclines towards the conduction axis, forming a constricted site (<6 Å) to prevent dehydrate/hydrate potassium ion (6–8 Å) flows (Fig. 5a). As such, the pre-open state1 is the closest to a closed state. In the pre-open state 2, the side chain of Y460s flips away from the conduction axis, extending the constricted site to wider than 8 Å (Fig. 5a), which may allow the dehydrated/hydrate potassium ion (6–8 Å) to flow at this constriction site. Consequently, the pre-open state 2 is closer to the conducting state. Therefore, the six sequential conformations, closed state 1, closed state 2, pre-open state1, pre-open state 2, open state, and pore dilation but non-conducting state constitute snapshots of hEag2 in transitions.

### A noncanonical ion conduction pathway
The canonical KcsA and MthK channels have an ion selectivity filter of the highly conserved signature sequence TVGYG, forming five to six ions occupied sites (S0–S4 or S0–S5)[7]. For hEag2, in a closed state, the four pore-lining S6 helices form a bundle-crossing in the middle of the membrane right below the ion selectivity filter with two layers of hydroxyphenyl or phenyl residue, Y460s and F464s (Supplementary Fig.10), forming the hydrophobic constriction sites through intrachain and interchain π–π interaction to prevent hydrated/dehydrated potassium ion flow (Fig. 5a, b). Below the hydrophobic constriction sites are hydrophilic constriction sites through intrachain and interchain electrostatic interaction of Q472s and T468s (Fig. 5c). The hydrophilic constriction sites together with the hydrophobic ones form a long channel gate (Fig. 2a–f). In the closed states and pre-open state 1, a strong and unambiguous density peak is located between the hydrophobic and hydrophilic constriction sites (Fig. 3a–c). The density is more likely to be a dehydrated potassium ion, and we name it site 6 or S6 as the seventh potassium ion selectivity site (Fig. 3a–c). Critically, both the pre-open state 2 and open state lose the S6 potassium ion, denoting a transition from closed to the open configuration during channel activation (Fig. 3d, e). Together, these results suggest that hEag2 has a noncanonical path for potassium conductivity and the S6 potassium ion may play an important role in channel activation.

### Iris-like rotation for hEag2 activation
The hydrophobic constriction sites (Y460s and F464s) form a bundle-crossing in the middle of the membrane that also divides the TM into two parts, the extracellular and the intracellular ion conduction pores (Supplementary Fig. 10). Structural comparison of all the states from closed states the pre-open states, the open state and the pore dilation but non-conducting state reveals that the extracellular ion conduction pore remains quite stable (Supplementary Fig. 11). Meanwhile, unlike the domain-swapped voltage-gated channels, which utilizing the proposed "move up and down" mechanism to sense the membrane voltage[8–10], the VSDs seem stereotypically stable without any appreciable conformational changes (Supplementary Fig. 12). Instead, the major conformational changes in response to voltage occur in the S4–S5 linker (Fig. 4a–d), leading to S5 moving towards the VSDs sequentially (Fig. 4e–i), accompanied by an iris-like rotation of S6 synchronically (Fig. 4e–i). Therefore, it consecutively extends the dilation of the intracellular pore from a closed to an open state (Figs. 4e–i and 5a–c). On the reverse direction from activation to resting state transition, the dilation of the intracellular pore is decreased in the pore dilation but non-conducting state and

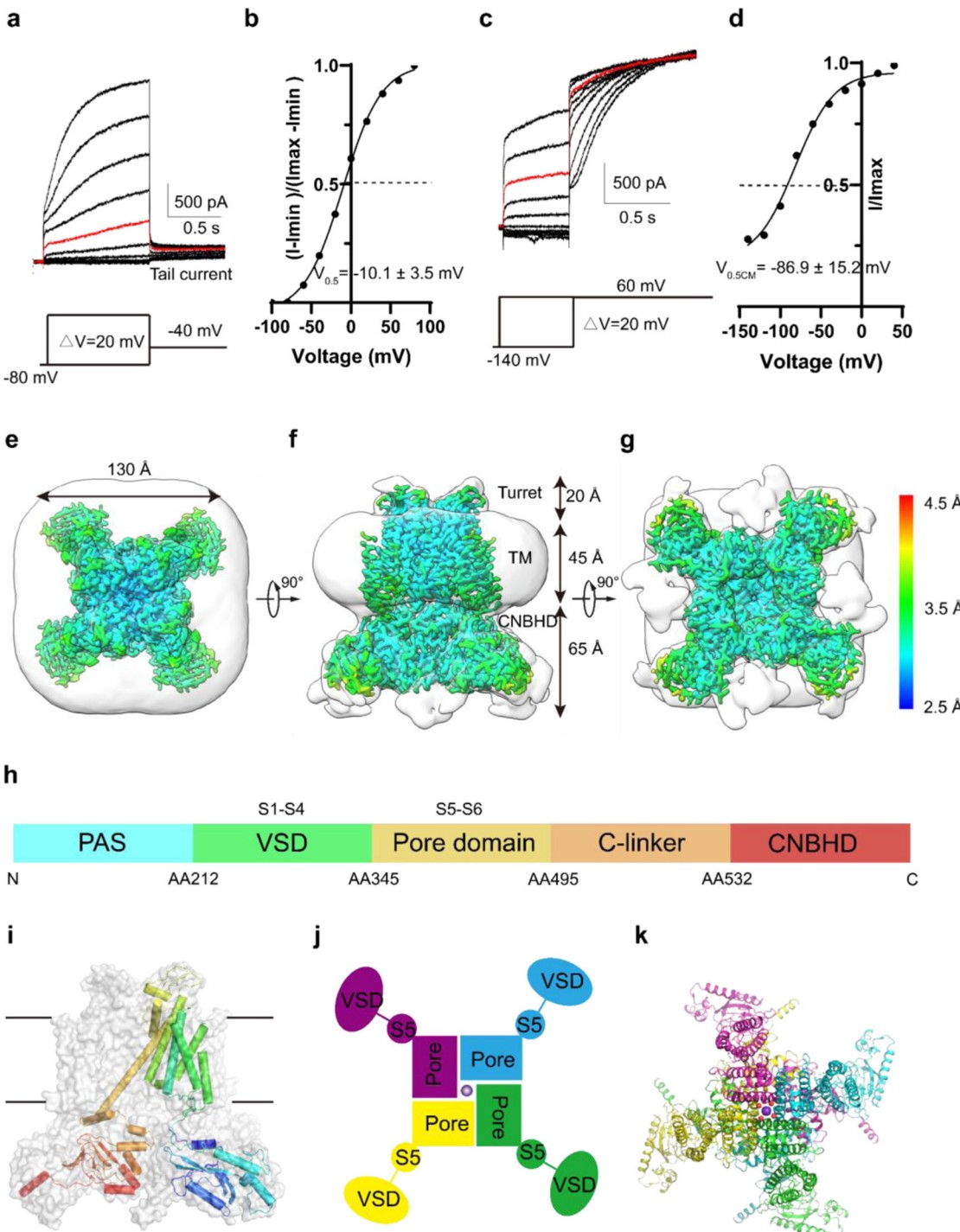

**Fig. 1 | Voltage-dependent activation and overall structure of hEag2.**
**a** Representative electrophysiological recording from the HEK293T cell expressing full-length hEag2 with the voltage-pulse protocol shown underlying. Tail currents used to generate activation curves are indicated. The red trace stands for the current recorded at the holding potential of 0 mV. **b** Normalized tail currents (($I$ $-I_{min}$)/($I_{max}-I_{min}$)) versus voltage ($I$–$V$ plot) from (**a**) was plotted and fit with a Boltzmann function to give a $V_{0.5}$ of −10.1 ± 3.5 mV (SEM). **c** Representative current traces demonstrating that the activation time of hEag2 increases after more hyperpolarized (negative) holding potentials. The voltage-pulse protocol is shown above recording. The red trace stands for the current recorded at the holding potential of 0 mV. **d** Plot of normalized Cole–Moore effect current of (**c**)

(Cole–Moore $I$–$V$ plot). The Cole–Moore $I$–$V$ plot was fit with a Boltzmann function to estimate the holding potential that produces half-maximal rates of activation. Top view (**e**), side view (**f**), and bottom view (**g**) of hEag2 overall structure with detergent micelle (gray) and estimated resolution are shown. **h** Schematic representation of hEag2 domain structure. Domains are rendered in colors according to the protomer structure shown in (**i**). **i** One subunit is represented as a cartoon and the remaining subunits are shown in surface representation. **j** Non-domain-swapped VSDs, pore domains and S5 of hEag2 tetramers in top view. A potassium ion in the pore domain is depicted as a purple sphere. **k** Overall structure of hEag2 tetramer in top view. Each subunit is rendered in colors.

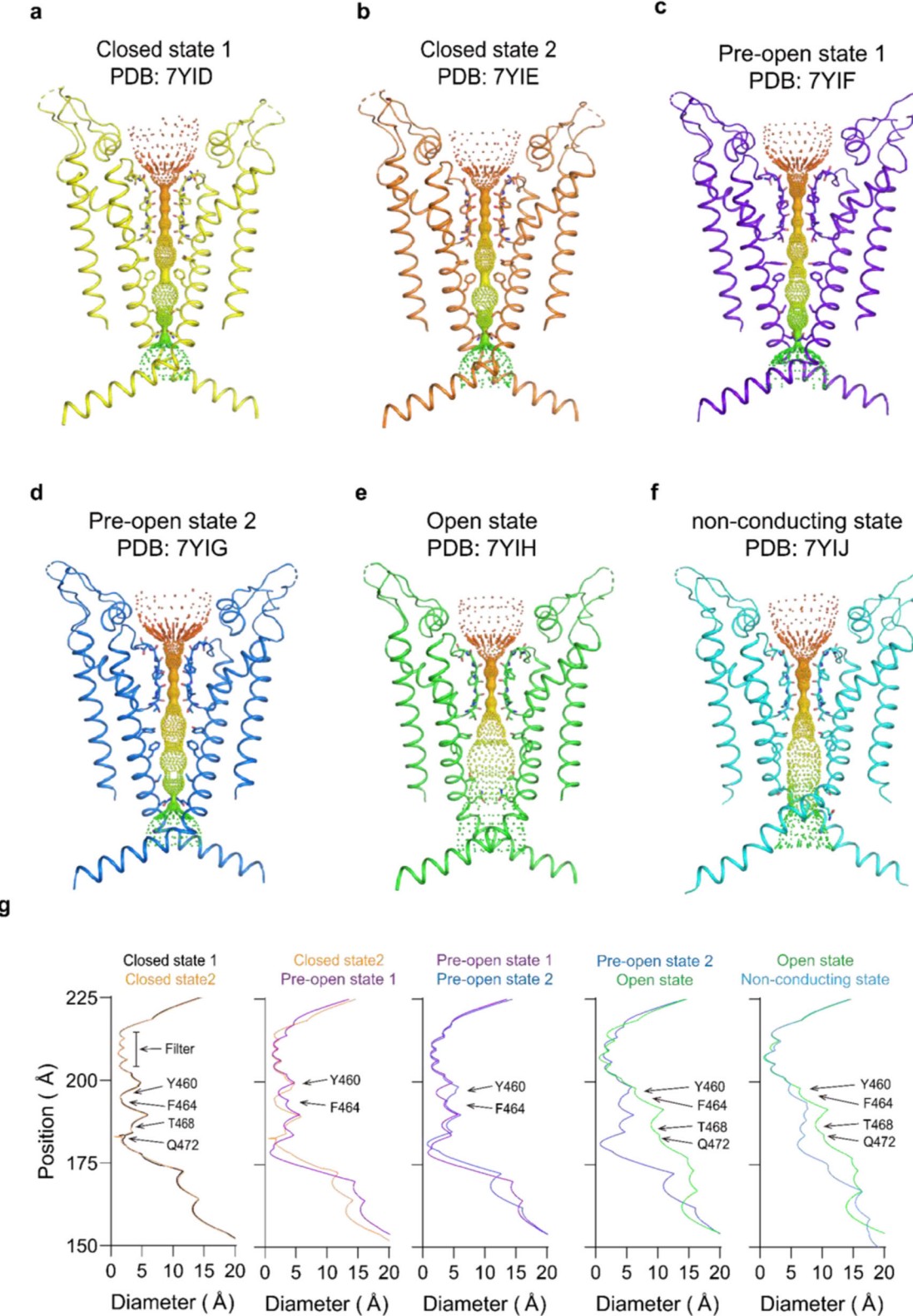

**Fig. 2 | Pore of hEag2 channel in six states. Ion pore with only two subunits of hEag2 in closed state 1. a–f** Closed state 1 (**a**); Closed state 2 (**b**); pre-open state 1 (**c**); pre-open state 2 (**d**); open state (**e**); and pore dilation but non-conducting state (**f**) shown, viewed from within the membrane. The minimal radial distance from the center axis to the protein surface is colored in iridescence. Selected residues facing the pore are in stick representation. **g** Comparison of the pore radius between two states of hEag2. The van der Waals radius is plotted against the distance along the pore axis. The selectivity filter and constricting residues are labeled.

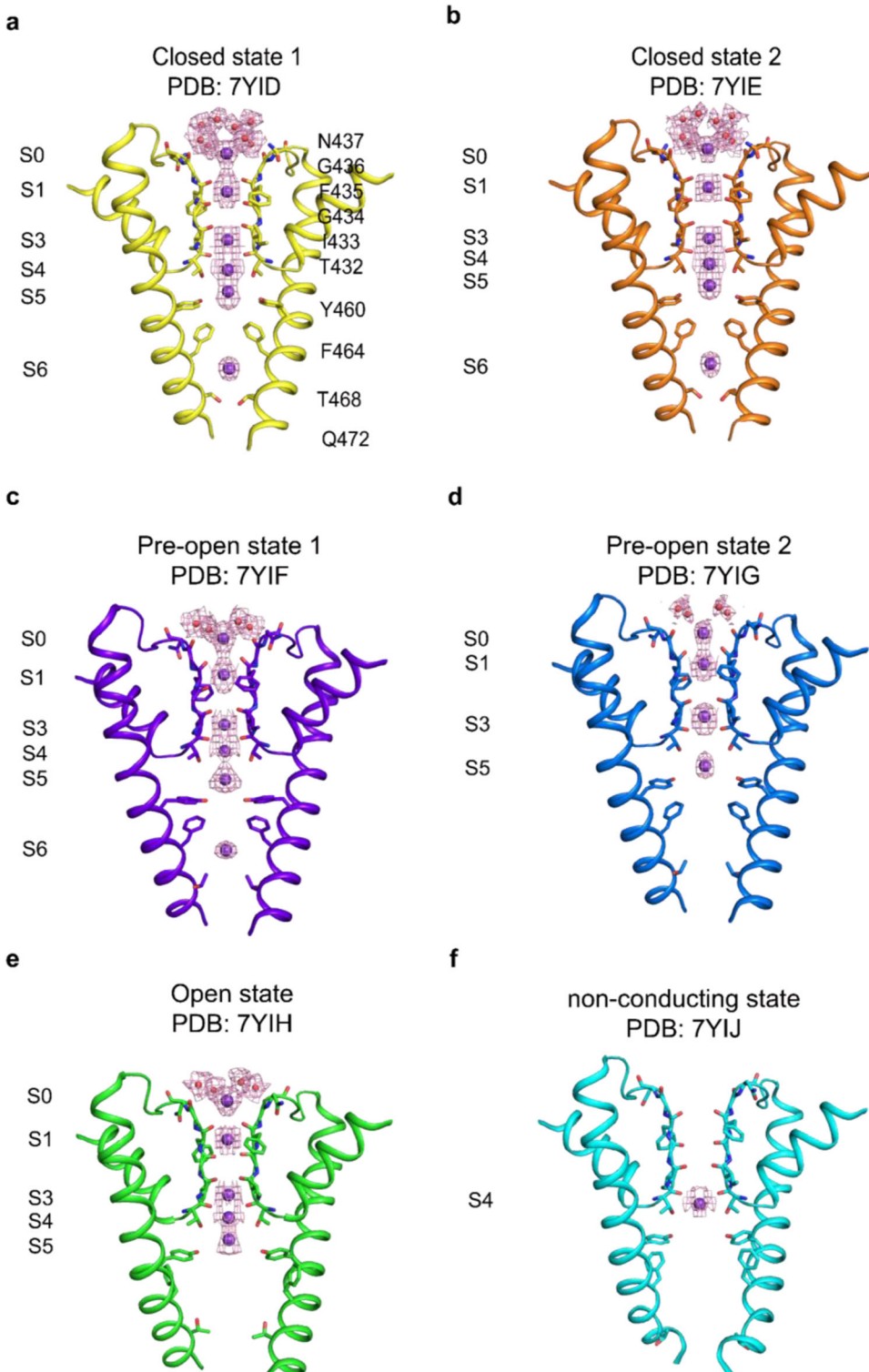

**Fig. 3 | Comparison of the selectivity filter and central cavity of hEag2 in six states. a–f** Selectivity filter and central cavity of hEag2 in closed state 1 in yellow (**a**), closed state 2 in orange (**b**), pre-open state 1 in slate (**c**), pre-open state 2 in blue (**d**), open state in green (**e**) and pore dilation but non-conducting state in cyan (**f**) shown, viewed from within the membrane. Water molecules above the selectivity filter are shown as red spheres and the potassium ions in S0–S6 are shown as purple spheres with electron density maps (purple mesh).

gradually transitions to the resting state. (Fig. 4g). Interestingly, the mutation K337A at the interface of the S4–S5 linker and S4 largely left-shift the half-activation membrane potential but has not impacted the steepness (zg) of the SS act curve (Fig. 4j, k), indicating that the conformational changes of S4–S5 linker may play a

critical role in voltage transducing. We also test the mutants of the typical K/R residues on S4, we found that they indeed play important role in channel voltage gating, indicating the VSDs may initiate the voltage sensing despite we do not observe the significant conformational changes of the VSDs (Supplementary Fig. 13, Table 2).

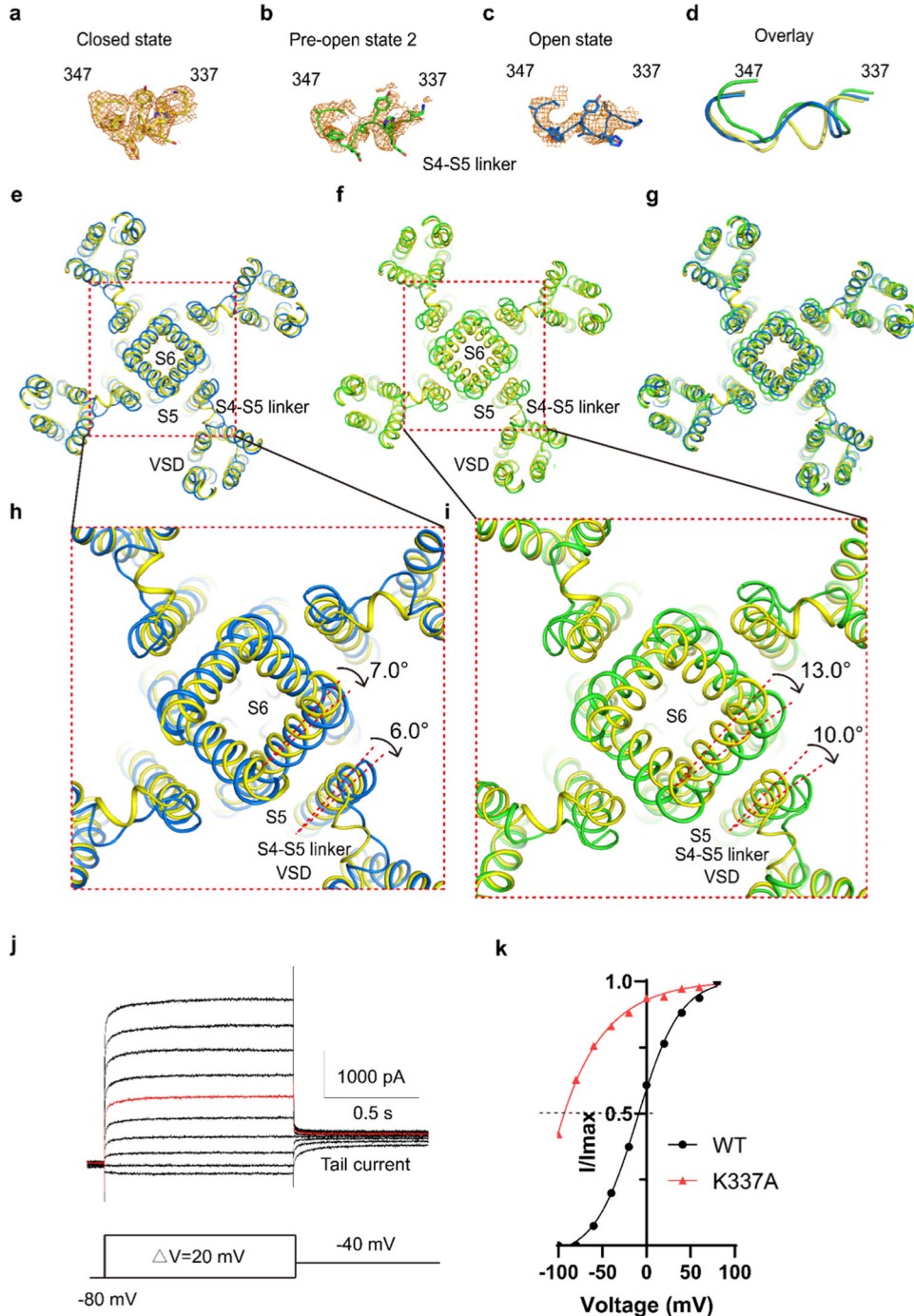

**Fig. 4 | Iris-like rotation of hEag2 pore domain and S5 during voltage activation.** **a–d** Structure comparison of S4–S5 linker (aa337–aa347) of hEag2 between the closed state (**a**), the pre-open state (**b**), and open state (**c**). **d** is an overlay view of the three states of the S4–S5 linker. **e** Structure comparison of pore domain, VSD and S4–S5 linker of hEag2 between the closed state and the pre-open state. **f** Structure comparison of pore domain, VSD and S4–S5 linker of hEag2 between the closed state and the open state. **g** Superposition of hEag2 pore domain, VSD and S4–S5 linker in the closed state, pre-open state, and open state. **h** Enlarged top view of extracellular pore domain and S5 of hEag2 in closed state and pre-open state.

**i** Enlarged top view of the pore domain and S5 of hEag2 in the closed state and pre-open state. The rotated direction and angle of the pore domain and S5 are labeled. Closed state is colored yellow, the pre-open state is colored slate, the open state is colored green and pore dilation but the non-conducting state is colored cyan. **j** Representative electrophysiological recording of K337A with the voltage-pulse protocol shown underlying. Tail currents used to generate activation curves are indicated. The red trace stands for the current recorded at the holding potential of 0 mV. **k** Normalized tail currents $I/I_{max}$ versus voltage ($I$–$V$ plot) from wild type and **j** were plotted and fit with a Boltzmann equation.

## Binary roles of F356s and Q472s in channel gating

The two constriction sites, F464s and Q472s, play binary roles in channel gating and stabilization. In the pre-open state, the F464s flip away from the ion conduction pore axis, breaking the interchain π–π interaction and extending the dilation of this constriction site to more than 6 Å (Fig. 5b) so that the dehydrated potassium ion can pass through. Meanwhile, noting that the side chain density of Q472s is weak in the pre-open state 2 (Fig. 5c), and the only constriction site for pre-open state 2 to break is the Q472s (Fig. 5a–c), if a voltage gradient is applied to the conduction pore at the Q472s and simply push the

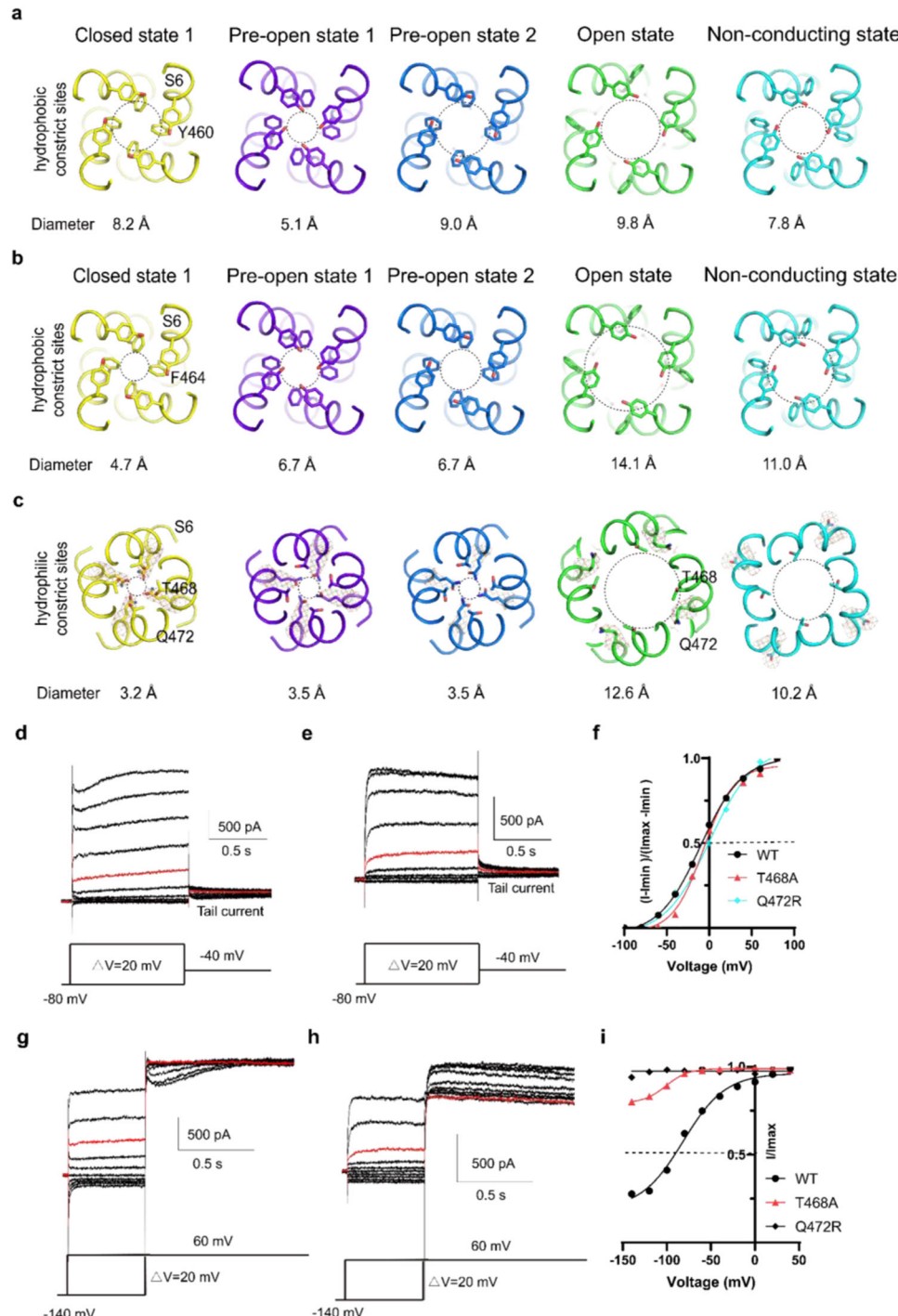

**Fig. 5 | Roles of T468s and Q472s in delayed rectified property and Cole–Moore effect.** Four hydrophobic constrict sites, Y460s (**a**), F464s (**b**), T468s, and Q472s (**c**), in the closed state (yellow), pre-open state 1 (slate), pre-open state 2 (blue), open state (green) and pore dilation but the non-conducting state (cyan) shown, viewed from the top. Each constricting site is depicted as a circle with a dotted line and the diameter of each circle is labeled underlying. The weak density of Q476 residue is shown as gray mesh. **d**–**e** Representative electrophysiological recording of T468A (**d**) and Q472R (**e**) with the voltage-pulse protocol shown underlying. Tail currents used to generate activation curves are indicated. The red trace stands for the current recorded at the holding potential of 0 mV. **f** Normalized tail currents $((I–I_{min})/(I_{max}–I_{min}))$ versus voltage ($I$–$V$ plot) from wild type and **d** and **e** were plotted and fit with a Boltzmann equation. **g**–**h** Representative current traces demonstrating that the activation time of T468A (**g**) and Q472R (**h**) increases after more hyperpolarized (negative) holding potentials. The voltage-pulse protocol is shown above recording. The red trace stands for the current recorded at the holding potential of 0 mV. **i** Plot of Cole–Moore $I$–$V$ plot from wild type, **g** and **h**. The Cole–Moore $I$–$V$ plot was fit with a Boltzmann equation to estimate the holding potential that produces half-maximal rates of activation.

dehydrate potassium ion into the ion occupied site of S6, the channel will assume a conducting state. On the other hand, in the open state, the F464s flip opposite of the ion conduction pore axis and towards S5, forming a new intrachain π–π interaction with F356 in S5 and may interact with V353 and the L458 at the adjacent S6 (Fig. 6c, d). In the meantime, the Q472s rotate towards the adjacent S6 and interact with the N466 at the adjacent S6 (Fig. 6c, d). Therefore, F464s and Q472s serve a dual role in gating and stabilizing hEag2 (Fig. 6a–d).

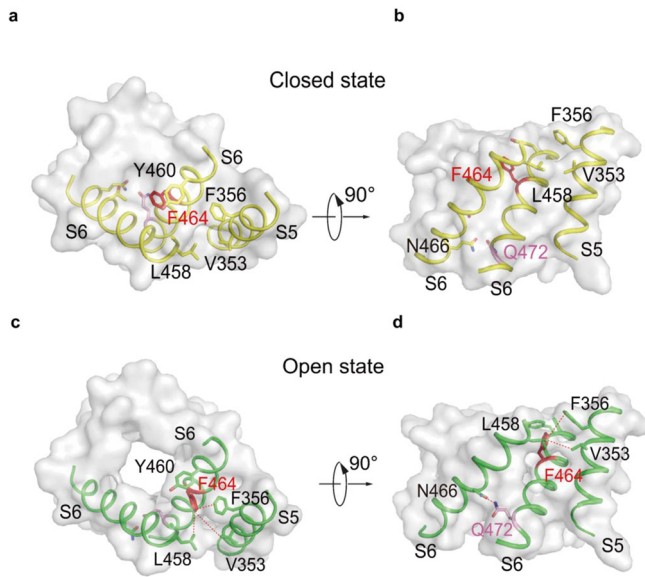

**Fig. 6 | Binary roles of F356s and Q472s in channel gating.** Cubic chart of hydrophobic constrict sites in hEag2 closed state, viewed from the top (**a**) and side (**b**). Cubic chart of hydrophobic constrict sites in hEag2 open state, viewed from the top (**c**) and side (**d**). S5 and S6 are represented as cartoons and the remaining residues are shown in surface representation. The constrict residues, F464 and Q472, are colored in red and pink, respectively. In the open state, F464 interacts with V353 and F356 on S5 and L458 on the adjacent S6, while Q472 interacts with N466 on the adjacent S6.

### T468s and Q472s confer delayed rectified property and Cole−Moore effect

The striking conformational changes of the ion-occupied site of S6 during resting to activation transitions suggest that it may play a critical role in delayed rectification and Cole−Moore effect (Figs. 2 and 3). To test this hypothesis, we generated a mutant T468A. The electrophysiological data reveals that T468As lessen the delayed rectify property and Cole−Moore effect (Fig. 5g, i), but have little effect on the voltage sensitivity (Fig. 5d, f). Additionally, the mutant Q472R almost eliminates the delayed rectify property and Cole−Moore effect (Fig. 5h, i), while the still has little effect on the voltage sensitivity (Fig. 5e, f). Interestingly, the corresponding mutant in hEag1 results in severe Temple−Baraitser syndrome (TBS) and epilepsy[11]. As the T468s and Q472s are the major gates of the S6 potassium ion, these results strongly indicate that the additional ion occupation site of S6 plays a vital role in delayed rectification property and Cole−Moore effect.

## Discussion

### Mechanism of voltage sensing and transducing of hEag2

Voltage acts as a simple and precious physical energy for cell signal communication. A gentle voltage change of either hyperpolarization or depolarization may largely change the conformational state of the voltage-gated ion channel and exchange the ion information[12]. The VSDs are proposed to be a maker of the voltage-gated ion channel which contains a lot of canonical voltage-gated ion channel families, such as the well-known Kv[13], Nav[14], Cav[15], etc. The VSD containing voltage-gated ion channels can also be divided into domain-swapped and domain-non-swapped subtypes[16]. The relatively well-studied domain-swapped voltage-gated ion channel senses the voltage by movement VSDs and transduces the electromechanical energy to the pore region through the long S4−S5 linker[17]. In the non-domain-swapped voltage-gated ion channel, the well-studied examples are the hyperpolarization-activated cyclic nucleotide-gated (HCN) channel. In the HCN channel, the VSDs bend towards the cell membrane for hyperpolarization response[18,19]. Our structures of hEag2 channels

reveal a voltage transducing mechanism, voltage energy induces the plasticity of the S4−S5 linkers then the S5 moves towards the VSDs sequentially, driving the intracellular pore region of S6 iris-like rotation synchronically. Therefore, it consecutively extends the dilation of the pore from a closed to an open state (Fig. 7, Supplementary Movie 1). These results of S4−S5 linker plasticity during the conformational changes are consistent with the recent observations of the Mackinnon group[20]. Meanwhile, some voltage-gated or voltage-dependent ion channels gated or tuned by voltage is lack of VSDs such as K2P[21] and TMEM16 channels[22]. The proposed mechanism of the non-canonical voltage-gated ion channel without the distinct VSDs is simply utilizing ion energy to break the ion conduction barriers[21,23]. Our pre-open state 2 is likely presenting a non-conducting to conducting transit state. If our closed states represent the deep closed state, we suspect that the voltage may act on both S4−S5 linkers and constricted sites for the hEag2 channel to sense the electromechanical force. However, there may exist a deep closed state (for example under −70 mV membrane potential condition), and our closed structures may still present a pre-open closed state. Therefore, it is plausible for an alternative hypothesis that hEag2 may still obey the VSD movement mechanism for voltage sensing, and the flexibility of the S4−S5 linker is only responsible for voltage transducing.

### Mechanism of delayed rectifier and Cole−Moore effects

The channel kinetics such as C-type inactivation, N-type inactivation, delayed rectifier activation, etc. may largely plastic the cellular ion homeostasis. The mechanism of C-type inactivation[24−26] and N-type inactivation[27,28] has some examples. However, the delayed rectifier activation coupled with the Cole−Moore effects remains elusive[3]. The delayed rectifier activation and Cole−Moore effects are proposed to be induced by multiple non-conducting states upon channel activation. Our four high-resolution non-conducting states give a glimpse of the channel transit from the deep closed state to the conducting state, presenting the majority of conformations of the delayed rectifier before the conducting state. As discussed above, the voltage may act on both S4−S5 linkers and constricted sites to allow ion flow. More importantly, we find the S6 potassium ion constrict sites, T468A and Q472R, significantly affect the delay rectifier property and the Cole−Moore effect without obvious voltage-sensitive changes. Therefore, we come up with the hypothesis that upon membrane depolarization, hEag2 transits the closed state to the pre-open state1 by breaking the energy barrier of the hydrophobic constrict site of F464s, and further electromechanical force drives the ion from the ion-occupied S6 to the S5. And if the continuous electrotechnical force drives the cytosolic ions into the constricted site of Q472s, the channel will tend to be conducting. Meanwhile, the more depolarization electrotechnical force applies to the ion conduction pore, the faster the channel current response to the voltage stimuli. Therefore, it may well explain the delayed rectifier and Cole−Moore effects.

## Methods

### Construct design

DNA fragments encoding hEag2 channel (UniProtKB Q8NCM2) were synthesized (Tsingke Biotechnology) and cloned into the plasmid Eric Gouaux (pEG) BacMam vector using EcoRI and XhoI restriction sites, where hEag2 gene is placed before a PreScission protease (Ppase) cleavage site, followed by a C-terminal enhanced green fluorescent protein (EGFP) and FLAG tag. The construct was identified by fluorescence detection size-exclusion chromatography (FSEC)[29].

### Protein expression and purification

The hEag2 was expressed in Expi293F cells using the BacMam method. Briefly, a bacmid carrying hEag2 was generated by transforming *E. coli* DH10Bac cells with the hEag2 construct according to the manufacturer's instructions (Bac-to-Bac; Invitrogen). Baculoviruses were

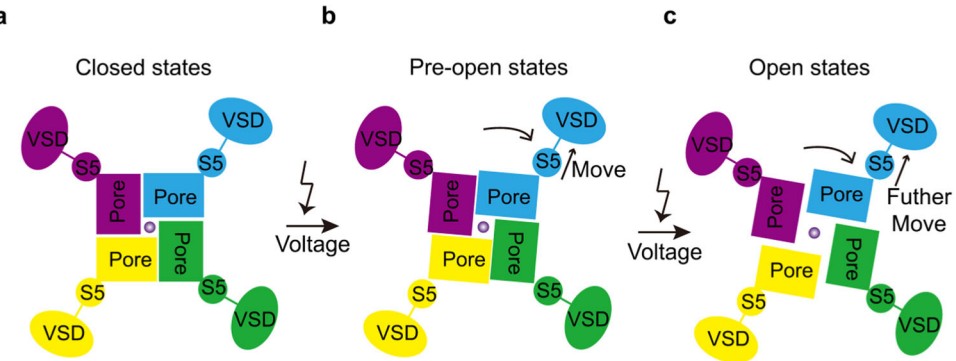

**Fig. 7 | Proposed voltage activation mechanism of hEag2 channel.** Voltage shock causes the movement of the S5 helix towards VSDs sequentially, which induces the pore domain open. The closed state (**a**), pre-open state (**b**), and open state (**c**) of the hEag2 channel viewed from the top is shown in the cartoon. Each subunit is colored and VSD, S5, and pore domain are labeled.

produced by transfecting Sf9 cells with the bacmid using Cellfectin II (Invitrogen). After three rounds of amplification, baculoviruses were used for cell transduction. Suspension cultures of Expi293F cells were grown at 37 °C to a density of ~3 × 10$^6$ cells/ml and viruses were added (8% v/v) to initiate the transduction. After 12 h, 10 mM sodium butyrate was supplemented, and the temperature was shifted to 30 °C. Cells were harvested at 60 h post-transduction.

The cell pellet from a 5 L culture was collected by centrifugation at 4000 r.p.m. and resuspended with 500 ml of lysis buffer (20 mM Tris–HCl, pH 8.0, 150 mM KCl), supplied with protease inhibitor cocktail containing 1.3 μg/ml aprotinin, 0.7 μg/ml pepstatin, and 5 μg/ml leupeptin and 2 mM phenylmethylsulfonyl fluoride (PMSF). Unbroken cells and cell debris were removed by centrifugation at 8,000 r.p.m. for 10 min. Supernatant was centrifuged at 36,000 r.p.m. for 30 min in a Ti45 rotor (Beckman). The membrane pellet was mechanically homogenized and solubilized in extraction buffer containing 20 mM Tris–HCl, pH 8.0, 150 mM KCl, 1% (w/v) lauryl maltose neopentyl glycol (LMNG), and 0.1% (w/v) cholesteryl hemisuccinate (CHS). Unsolubilized materials were removed by centrifugation at 36,000 r.p.m. for 1 h in a Ti45 rotor. Supernatant was loaded onto anti-FLAG G1 affinity resin (Genscript) by gravity flow. Resin was further washed with 10 column volumes of wash buffer (lysis buffer with 0.02% LMNG), and protein was eluted with an elution buffer (lysis buffer with 0.02% LMNG and 230 μg/ml FLAG peptide). The C-terminal GFP tag of eluted protein was removed by human rhinovirus 3C (HRV 3C) protease cleavage for 2 h. The protein was further concentrated by a 100-kDa cutoff concentrator (Milipore) and loaded onto a Superose 6 increase 10/300 column (GE Healthcare) running in lysis buffer with 1 mM MgCl$_2$ and 0.03% digitonin. Peak fractions were pooled and concentrated to around 5 mg/ml for cryo-EM sample preparation. All steps were performed at 4 °C.

### Cryo-EM sample preparation
For cryo-EM sample preparation, aliquots (3 μl) of the protein sample were loaded onto glow-discharged (20 s, 15 mA; Pelco easiGlow, Ted Pella) Au grids (Quantifoil, R1.2/1.3, 300 mesh). The grids were blotted for 5 s with 3 force after waiting for 5 s and immersed in liquid ethane using Vitrobot (Mark IV, Thermo Fisher Scientific/FEI) in conditions of 100% humidity and 8 °C.

### Cryo-EM image acquisition
Cryo-EM data for hEag2 were collected on a Titan Krios microscope (FEI) equipped with a cesium corrector operated at 300 kV. Movie stacks were automatically acquired with EPU software[30] on a Gatan K3 Summit detector in super-resolution mode (×105,000 magnification) with pixel size 0.4245 Å at the object plane and with defocus ranging from −1.5 to −2.0 μm and GIF Quantum energy filter with a 20 eV slit

width. The dose rate on the sample was 23.438 e$^-$ s$^{-1}$ Å$^{-2}$, and each stack was 2.56 s long and dose-fractioned into 32 frames with 80 ms for each frame. Total exposure was 60 e$^-$ Å$^{-2}$. Acquisition parameters are summarized in Table S1.

### Image processing
A flowchart of the Cryo-EM data processing process can be found in Fig. S2. Data processing was carried out with the cryoSPARC v3.2.0 suite[31]. Super-resolution image stacks were gain-normalized, binned by 2 with Fourier cropping, and patch-based CTF parameters of the dose-weighted micrographs (0.849 Å per pixel) were determined by cryoSPARC. Around 475K particles were picked by blob picking from 999 micrographs and 2D classification was performed to select 2D classes as templates. Template picker was used to pick 868,893 particles. Particles were extracted using a 400-pixel box with a pixel size of 0.849 Å. The dataset was cleared using several rounds of 2D classification, followed by Ab initio reconstruction using C1 symmetry and initial model (PDB: 6PBY). Two classes with 638K particles (82.3%) were further classified into eight classes using another round of Ab initio reconstruction. Six maps were refined using non-uniform refinement and local refinement for reconstructing the density map while imposing a C4 symmetry.

The six final maps were low-pass filtered to the map-model FSC value. The reported resolutions were based on the FSC = 0.143 criterion.

### Model building
The atomic models of monomers were built in Coot based on an initial model (PDB: 6BPY). The models were then manually adjusted in Coot. Tetramer models were obtained subsequently by applying a symmetry operation on the monomer. These tetramer models were refined using phenix.real_space_refine with secondary structure restraints by phenix and Coot iteratively. Residues whose side chains have poor density were modeled as alanines. For validation, FSC curves were calculated between the final models and EM maps. The pore radii were calculated using HOLE. Figures were prepared using PyMOL and Chimera.

### Electrophysiological recording
HEK293T cells were cultured on coverslips placed in a 12-well plate containing DMEM/F12 medium (Gibco) supplemented with 10% fetal bovine serum (FBS). The cells in each well were transiently transfected with 1 μg hEag2 DNA plasmid using Lipofectamine 3000 (Invitrogen) according to the manufacturer's instructions. After 12–20 h, the coverslips were transferred to a recording chamber containing the external solution (10 mM HEPES-Na pH 7.4, 150 mM NaCl, 5 mM glucose, 2 mM MgCl$_2$, and 1 mM CaCl$_2$). Borosilicate micropipettes (OD 1.5 mm, ID 0.86 mm, Sutter) were pulled and fire polished to 2–5 MΩ

resistance. For inside-out recordings, the pipette solution was 10 mM HEPES-Na pH 7.4, 150 mM KCl, and 5 mM EGTA. The bath solution was 10 mM HEPES-Na pH 7.4, 150 mM NaCl, 5 mM glucose, 2 mM $MgCl_2$, and 1 mM $CaCl_2$.

For the GUVs patch clamp analysis, the giant unilamellar vesicles (GUVs) reconstituted with the purified hEag2 (protein:lipid = 1:1000, wt:wt; azolectin (Sigma)) were prepared by a modified sucrose method. First, 250 µl of a 20 mg/ml solution of azolectin in chloroform was dried in a glass bottle under a stream of $N_2$ while rotating the bottle, to produce a homogeneous dried lipid film. Subsequently, 250 µl of 0.4 M sucrose was placed at the bottom of the bottle, and the solution was incubated at 55 °C for 2–3 h until the lipid was resuspended. After cooling the solution to 4 °C, the purified proteins were added to achieve the desired protein-to-lipid ratio. The glass bottle containing the protein–lipid solution was shaken gently on an orbital mixer for 3 h at 4 °C. Approximately 50 mg blotted dry Biobeads-SM2 adsorbents (washed by rotating in methanol for 30 min three times, water for 30 min three times, and DR buffer for 30 min three times) was then added and the reconstitution was rotated at 4 °C for 3 additional hours. After this procedure, the sample was ready for patch clamping. The pipette buffer contained 5 mM Hepes, 180 mM NaCl, 20 mM KCl, pH 7.2 (adjusted with NaOH) and the bath solution was 5 mM Hepes, 200 mM KCl, 40 mM $MgCl_2$, pH 7.2 (adjusted with KOH).

Recordings were obtained at room temperature (-25 °C) using an Axopatch 200B amplifier, a Digidata 1550 digitizer, and pCLAMP 10.7 software (Molecular Devices). The patches were held at −80 mV and the recordings were low-pass filtered at 1 kHz and sampled at 20 kHz. Statistical analyses were performed using GraphPad Prism 9.

### Reporting summary
Further information on research design is available in the Nature Portfolio Reporting Summary linked to this article.

## Data availability
The data that support this study are available from the corresponding authors upon request. Cryo-EM maps have been deposited in the Electron Microscopy Data Base (EMDB) under accession codes EMD-33855 (closed state 1), EMD-33856 (closed state 2), EMD-33857 (pre-open state 1), EMD-33858 (pre-open state 2), EMD-33859 (open state), and EMD-33860 (pore dilation non-conducting state). The coordinates have been deposited in the Protein Data Bank (PDB) under accession codes 7YID (closed state 1), 7YIE (closed state 2), 7YIF (pre-open state 1), 7YIG (pre-open state 2), 7YIH (open state), and 7YIJ (pore dilation non-conducting state). The rEAG1 structure used in this study can be accessed using accession code 6PBY. Source data are provided with this paper.

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

## Acknowledgements

We would like to thank the Cryo-EM Facility and High-Performance Computing (HPC) Center of Westlake University for providing cryo-EM and computation support. This work was supported by Westlake Laboratory (Westlake Laboratory of Life Sciences and Biomedicine) and an Institutional Startup Grant from the Westlake Education Foundation to D.P. We also would like to thank all the Cell fate control lab members for their support.

## Author contributions

M.Z. and D.P. conceived the project. Y.S. and M.Z. designed the experiments. Y.S. performed electrophysiology experiments and prepared the cryo-EM sample. Y.S. and M.Z. collected cryo-EM data. M.Z. performed image processing and analyzed EM data. Y.S. and M.Z. built the model and wrote the manuscript draft. All authors contributed to the manuscript preparation.

## Competing interests

The authors declare no competing interests.
