## [Peer Review File · Nature Communications]

Mechanism underlying delayed rectifying in human voltage-mediated activation Eag2 channelReviewers' Comments:

Reviewer #1:

Remarks to the Author:

In this study, Zhang and colleagues have determined 6 near-atomic resolution structures of hEAG2 channels using cryo-EM. Classification of the conformational states, into closed, pre-open, open and inactivated states is based on the pore dilation and ion distribution.

Based on this data, supported to some extent by functional data, they make three claims. 1. That F464 and Q472 play critical roles in mediating the Cole-Moore shift in these channels. 2. that changes in K⁺ occupancy of the selectivity filter mediate inactivation and 3. that the voltage-sensitivity for activation of non-domain swapped K⁺ channels utilise a novel mechanism involving movement of the S4S5 linker and the S5 helix relative to the pore domain.

The data in support of a role for F464 and Q472 in mediating the Cole-Moore shift is strong. There are clear movements of these side chains seen in the cryoEM densities as well as the models and validated using site-directed mutagenesis and electrophysiology data. This is also consistent with previous literature suggesting that the S6 region was important for the Cole-Moore shift seen in HEAG, a close homologue.

The conclusions they deduce re: the mechanism of voltage-dependent activation and "inactivation" however are less robust, and not well supported by functional data.

The idea that changes to K⁺ occupancy affect inactivation is not novel for K⁺ channels. Furthermore, there is not strong evidence that HEAG2 channels inactivate to any appreciable degree, so the significance of these findings are uncertain. Additionally, given the difficulties in determining whether densities on the central axis represent water and/or K⁺ ions at the resolution of these structures (3.4-3.8Å) these findings should be interpreted more cautiously.

The authors claim that they have identified a novel mechanism for voltage sensing (movement of S4S5 linker and associated re-orientation of S5 with respect to the pore domain) is potentially the most exciting part of the study but also the most speculative and requires much further investigation. The authors acknowledge that there is no significant difference in the structures of the voltage sensor domains in any of the structures. In the absence of translocation of a positive charge from the VSD (between closed and activated states) one needs to consider what then are the charge(s) that are moving (positive up or negative down) in response to the change in membrane potential to account for the voltage sensitivity. Identifying these charge(s) to confirm that they have indeed identified a novel voltage sensing mechanism is critical. The authors claim that the mutation of K337 (on the S4S5 linker) increases the voltage sensitivity is not correct. Voltage sensitivity is related to the steepness (zg) of the SS act curve, not the voltage range over which activation takes place, which reflects the dG0 of channel activation. The slopes of the two lines in Figure 4k look similar. Thus K337 is not a strong candidate for being the critical charge carrier. I also have concerns about the interpretation of the movement of S4S5 linker. For example, in the fit of 7YIJ to its corresponding map, almost all of the residues in the linker K337-A347 are classified as having a poor fit to the cryoEM map. Figure 4a,b,c which shows movement of the S4S5 linkers between states, would be much more convincing if the matching cryoEM density was also shown as it would convince the readers that this movement is real. I note that the authors have done this for the T468/Q472 sidechains in Figure 5c and this makes the data in that figure much more convincing.

An alternative hypothesis is that they have not captured a fully closed state but rather a pre-open closed state (voltage sensors have moved up but the pores have not yet opened). In the absence of evidence to the contrary I see this as the most plausible explanation for their data.

Overall the methodology is sound and there is sufficient detail in the methods to enable others to replicate the work.

Minor points

If the whole map was solved as a single class, what was the overall resolution. If the increased number of particles included in the analysis did not improve resolution compared to the 6 separate classes then one would be more confident that there was significant conformational heterogeneity within the samples and strengthen confidence in the different states identified in the separate classes

of particles.

In extended Data Figure 8, 5VA3 is HERG S631A – which is an open state structure not a closed state structure

In extended Data Figure 12 what are the isolated small straight lines in panels a-f. Is there any significance to the observation that the closed and inactivated states are more similar to each other than to the open state or pre-open states.

On line 177, should R462R be Q472R

Reviewer #2:

Remarks to the Author:

The paper of Zhang, Shan and Pei is from a structural point of view a putative description of two novel mechanisms for ion permeation and voltage sensitivity in the hEag2 channel. The authors describe six (6) different conformations of the hEag2 channel that putatively depict the gating activation/deactivation and perhaps inactivation gating of this channels. The 6 structures portrait very little changes of the four voltage sensing domains, despite the channel's conformations were obtained at 0 mV, instead the major conformational change occur in the S4-S5 linker, which leads to an outward movement of the S5 segment that in turn also allows an iris-like rotation of S6 synchronically. Additionally, these novel hEag2 channel structures display an extra K⁺ in a putative intracellular binding site, they called named S6, beside the canonical 5 K⁺ found in other potassium channels. The K⁺ bound to the S6 binding site is dehydrated and the authors based on their structural interpretations and in electrophysiological data of the T468As postulated that conformational changes at this binding site play a critical role in delayed rectification and Cole-Moore effect displayed by this channel. Finally, there was a putative "inactive" hEag2 conformation that accordingly was trapped in the open-inactivated state. The authors proposed that this conformation correspond to an inactivated one given that the channel shows only one ion at the S4 binding site of the selectivity filter.

Major concerns:

- 1) It is very concerning for this reviewer the biochemical quality of the purified hEag2. The size exclusion chromatography (SEC) profile showed no discernible monodispersed peak and the protein gel pooled for structural determination showed several bands that the authors do not explain. So, this could be highly degraded hEag2 channels that obviously put in doubt the author's structural interpretations.
- 2) The functional state of the hEag2 biochemical preparation is unknown and giving the low quality of the biochemical preparation, the authors should provide some functional validation.

Point-by-Point Response to Reviewer's Comments

Reviewer #1 (Remarks to the Author):

In this study, Zhang and colleagues have determined 6 near-atomic resolution structures of hEAG2 channels using cryo-EM. Classification of the conformational states, into closed, pre-open, open and inactivated states is based on the pore dilation and ion distribution.

Based on this data, supported to some extent by functional data, they make three claims. 1. That F464 and Q472 play critical roles in mediating the Cole-Moore shift in these channels. 2. that changes in K⁺ occupancy of the selectivity filter mediate inactivation and 3. that the voltage-sensitivity for activation of non-domain swapped K⁺ channels utilise a novel mechanism involving movement of the S4S5 linker and the S5 helix relative to the pore domain.

The data in support of a role for F464 and Q472 in mediating the Cole-Moore shift is strong. There are clear movements of these side chains seen in the cryoEM densities as well as the models and validated using site-directed mutagenesis and electrophysiology data. This is also consistent with previous literature suggesting that the S6 region was important for the Cole-Moore shift seen in HEAG, a close homologue.

We appreciate this positive comment from reviewer #1.

The conclusions they deduce re: the mechanism of voltage-dependent activation and “inactivation” however are less robust, and not well supported by functional data.

The idea that changes to K⁺ occupancy affect inactivation is not novel for K⁺ channels. Furthermore, there is not strong evidence that HEAG2 channels inactivate to any appreciable degree, so the significance of these findings are uncertain. Additionally, given the difficulties in determining whether densities on the central axis represent water and/or K⁺ ions at the resolution of these structures (3.4-3.8 Å) these findings should be interpreted more cautiously.

We also appreciate these comments as well. Indeed, we noticed that our electrophysiological data of hEag2 channel show no significant inactivation, so the statement of “inactivation” state (such as C-type inactivation state) is not suitable. As hEag2 channel as other channels will undergo close to open and then transit to closed state (which forms basis of the single channel signal), the “inactivation” state may stand for the open to closed transit state. We appreciate the reviewer's suggestion and therefore revised the statement of “inactivation” state to re-closed state.

The authors claim that they have identified a novel mechanism for voltage sensing (movement of S4S5 linker and associated re-orientation of S5 with respect to the pore domain) is potentially the most exciting part of the study but also the most speculative and requires much further investigation. The authors acknowledge that there is no significant difference in the structures of the voltage sensor domains in any of the structures. In the absence of translocation of a positive charge from the VSD (between closed and activated states) one needs to consider what then are the charge(s) that are moving (positive up or negative down) in response to the change in membrane potential to account for the voltage

sensitivity. Identifying these charge(s) to confirm that they have indeed identified a novel voltage sensing mechanism is critical. The authors claim that the mutation of K337 (on the S4S5 linker) increases the voltage sensitivity is not correct. Voltage sensitivity is related to the steepness (zg) of the SS act curve, not the voltage range over which activation takes place, which reflects the dG0 of channel activation. The slopes of the two lines in Figure 4k look similar. Thus K337 is not a strong candidate for being the critical charge carrier. I also have concerns about the interpretation of the movement of S4S5 linker. For example, in the fit of 7YIJ to its corresponding map, almost all of the residues in the linker K337-A347 are classified as having a poor fit to the cryoEM map. Figure 4a,b,c which shows movement of the S4S5 linkers between states, would be much more convincing if the matching cryoEM density was also shown as it would convince the readers that this movement is real. I note that the authors have done this for the T468/Q472 sidechains in Figure 5c and this makes the data in that figure much more convincing.

An alternative hypothesis is that they have not captured a fully closed state but rather a pre-open closed state (voltage sensors have moved up but the pores have not yet opened). In the absence of evidence to the contrary I see this as the most plausible explanation for their data.

We agree with review #1's suggestion. We added the cryo-EM density to Figure 4a, b, c.

We also tested the alternative hypothesis by performing mutagenesis of the potential key residues on VSD region and found they indeed play important role in channel voltage gating and thus added the alternative voltage gating hypothesis in the discussion.

Fig. 1. Representative I-V curve from the HEK293T cell expressing WT hEag2 and its mutation at VSD region. The results indicate that mutating at VSD domain impair channel voltage sensitive, resulting the LOF phenotype.

Minor points

If the whole map was solved as a single class, what was the overall resolution. If the increased number of particles included in the analysis did not improve resolution compared to the 6 separate classes then one would be more confident that there was significant conformational heterogeneity within the samples and strengthen confidence in the different states identified in the separate classes of particles.

We thank the review #1 suggestion. We used all six classes' particles to perform refinement and got a worse map (3.5 Å lower than the closed map of 3.4 Å), suggesting there was significant conformational heterogeneity within the samples.

In extended Data Figure 8, 5VA3 is HERG S631A – which is an open state structure not a closed state structure

We made the revision in Extended Data Figure 8.

In extended Data Figure 12 what are the isolated small straight lines in panels a-f. Is there any significance to the observation that the closed and inactivated states are more similar to each other than to the open state or pre-open states.

We apologize for the confusion generated by the extended Data Figure 12. The straight lines in panels a-f are flexible loops between S1 and S2. These flexible loops with poor density were not built during the model building and shown as dash lines. Therefore, the straight lines have no meaning for structure comparison.

On line 177, should R462R be Q472R

We made the revision on line 177.

Reviewer #2 (Remarks to the Author):

The paper of Zhang, Shan and Pei is from a structural point of view a putative description of two novel mechanisms for ion permeation and voltage sensitivity in the hEag2 channel. The authors describe six (6) different conformations of the hEag2 channel that putatively depict the gating activation/deactivation and perhaps inactivation gating of this channels. The 6 structures portrait very little changes of the four voltage sensing domains, despite the channel's conformations were obtained at 0 mV, instead the major conformational change occur in the S4-S5 linker, which leads to an outward movement of the S5 segment that in turn also allows an iris-like rotation of S6 synchronically. Additionally, these novel hEaG2 channel structures display an extra K⁺ in a putative intracellular binding site, they called named S6, beside the canonical 5 K⁺ found in other potassium channels. The K⁺ bound to the S6 binding site is dehydrated and the authors based on their structural interpretations and in electrophysiological data of the T468As postulated that conformational changes at this binding site play a critical role in delayed rectification and Cole-Moore effect displayed by this channel. Finally, there was a putative “inactive” hEag2 conformation that accordingly was trapped in the open-inactivated state. The authors proposed that this conformation correspond to an inactivated one given that the channel shows only one ion at the S4 binding site of the selectivity filter.

Major concerns:

1) It is very concerning for this reviewer the biochemical quality of the purified

hEag2. The size exclusion chromatography (SEC) profile showed no discernible monodispersed peak and the protein gel pooled for structural determination showed several bands that the authors do not explain. So, this could be highly degraded hEag2 channels that obviously put in doubt the author's structural interpretations.

We thank reviewer #2's constructive comments. The absence of no discernible monodispersed peak may be the results of binding partners with Eag2 (For example the endogenous calmodulin). The closed maps show weak density of calmodulin in the corresponding region of the calmodulin and rEag1 complex. Meanwhile, the multiple bands represent the PTM (Such as glycosylation) of hEag2 commonly existing in the membrane proteins. Indeed, glycosylation density also can be observed in our maps.

Fig. 2. Glycosylation density of hEAG2 at N403.

2) The functional state of the hEag2 biochemical preparation is unknown and giving the low quality of the biochemical preparation, the authors should provide some functional validation.

We appreciate this interesting suggestion. Now, we have reconstituted the purified hEag2 which was used for cryo-EM into the giant unilamellar vesicles (GUVs) and recorded the typical voltage gated rectified currents to provide the functional validation.

Thanks again all reviewers for putting so much effort into reviewing our paper.

Sincerely,
Yours.

Reviewers' Comments:

Reviewer #1:

Remarks to the Author:

The authors have partially addressed the two main concerns I had with the earlier version of the manuscript.

1. Mechanism of Voltage sensing. The authors have provided additional data (but only in the response to reviewers not in the manuscript) suggesting that the VSD likely contributes to voltage sensing. Intriguingly, the data in the response to reviewers is only for mutants in the cytoplasmic S2-S3 region within the VSD, not the classical voltage sensor (i.e. S4 region). Based on this data the authors have modified their manuscript to indicate that movement of the S4S5 linker is likely important for transduction of voltage sensing to pore opening rather than being intrinsic for voltage sensing per se. I agree that this is likely correct, however they could strengthen the manuscript by including the new EP data in the paper (a supplementary figure would be fine) and expand this to include some S4 mutants (e.g. K324, R327, R330, R333). The authors should also include a table that summarises the DG0 and zg (from limiting slope analysis) for the K337A as well as other mutants they have studied. I note that the new interpretation of the role of the S4S5 linker is consistent with that proposed by the Mackinnon group in their recent paper (PMID: 36331999). I appreciate the new Mackinnon paper was published after this manuscript was submitted but it would be worth revising the discussion to include reference to the new Mackinnon paper
2. Inactivation. The authors have changed their wording in the manuscript to "re-closed" instead of inactivation. I suggest using non-conducting state as a more generic term as the intracellular pore gate is still in the open conformation.

Reviewer #2:

None

Point-by-Point Response to Reviewer's Comments

Thank you again for submitting your revised manuscript "Voltage-mediated activation of human oncogenic and epileptogenic Eag2 channel" to Nature Communications. We have now received reports from the reviewers who evaluated the original version. On the basis of their comments (copied below), we have decided to invite an additional revision of your work.

You will see that, while the reviewers find that your revisions improved the manuscript, some important points remain to be addressed. Specifically, as indicated by reviewer #1, we would like to see a more thorough analysis of the electrophysiology from the mutant constructs that have been studied.

We also note that in your reply to reviewer #2 about the biochemical purity of your samples, glycosylation or other PTM would account for bands at a higher molecular weight than what is expected for a single hEag2 subunit. However it is clear from the SDS-PAGE gel that there are several bands below the expected molecular weight. A revised version of the manuscript will need to reconcile by some bands are lower than expected. Please revise your manuscript, addressing all the remaining issues raised by the reviewers.

We apologize for forgetting labeling the potential HSP70 protein of the bands lower than expected. We marked it in the supplementary figure.

REVIEWER COMMENTS

Reviewer #1 (Remarks to the Author):

The authors have partially addressed the two main concerns I had with the earlier version of the manuscript.

1. Mechanism of Voltage sensing. The authors have provided additional data (but only in the response to reviewers not in the manuscript) suggesting that the VSD likely contributes to voltage sensing. Intriguingly, the data in the response to reviewers is only for mutants in the cytoplasmic S2-S3 region within the VSD, not the classical voltage sensor (i.e. S4 region). Based on this data the authors have modified their manuscript to indicate that movement of the S4S5 linker is likely important for transduction of voltage sensing to pore opening rather than being intrinsic for voltage sensing per se. I agree that this is likely correct, however they could strengthen the manuscript by including the new EP data in the paper (a supplementary figure would be fine) and expand this to include some S4 mutants (e.g. K324, R327, R330, R333). The authors should also include a table that summarises the DG0 and zg (from limiting slope analysis) for the K337A as well as other mutants they have studied.

I note that the new interpretation of the role of the S4S5 linker is consistent with that proposed by the Mackinnon group in their recent paper (PMID: 36331999). I appreciate the new Mackinnon paper was published after this manuscript was submitted but it would be worth revising the discussion to include reference to the new Mackinnon paper

We appreciated the review #1 suggestion. We performed the EP experiment of key mutants on S4 and add them as a supplementary figure and a table that summarizes the DG0 and zg for all mutants in the paper. We also discussed the new Mackinnon paper in the discussion.

2. Inactivation. The authors have changed their wording in the manuscript to "re-closed" instead of inactivation. I suggest using non-conducting state as a more generic term as the intracellular pore gate is still in the open conformation.

We thank the review #1 constructive suggestion. We used pore dilation but non-conducting state in the paper.

Reviewers' Comments:

Reviewer #1:

Remarks to the Author:

The authors have addressed the concerns that I raised in my previous review. However, there are some errors in supplementary Table 2. Z_g is inversely related to k and so as k goes up, Z_g should go down. Based on the $V_{0.5}$ and k values given the dG_0 values are correct but the Z_g values should be: WT (0.45), T468A (0.61), Q472R (0.42), K337A (0.41), R330A (0.33), R333A (0.83).

Furthermore for the data in supplementary Table 2, there should be errors given and values should be presented with an appropriate number of significant figures. For example the $V_{0.5}$ for WT should be given as 56 rather than 56.14823133.

REVIEWERS' COMMENTS

Reviewer #1 (Remarks to the Author):

The authors have addressed the concerns that I raised in my previous review. However, there are some errors in supplementary Table 2. Zg is inversely related to k and so as k goes up, zg should go down. Based on the V0.5 and k values given the dG0 values are correct but the zg values should be: WT (0.45), T468A (0.61), Q472R (0.42), K337A (0.41), R330A (0.33), R333A (0.83). Furthermore for the data in supplementary Table 2, there should be errors given and values should be presented with an appropriate number of significant figures. For example the V0.5 for WT should be given as 56 rather than 56.14823133.

We revised the table as the reviewer #1' suggestion.